# General psychiatric inpatient care for persons with anorexia nervosa—an integrative literature review protocol

Anna Sandsten [1], Britt-Marie Lindgren,[2] Maria Strömbäck [3], Git-Marie Ejneborn-Looi,[4] Sebastian Gabrielsson [4]

¹Department of Nursing, Umea universitet Medicinska fakulteten, Umea, Sweden
²Department of Nursing, Umeå universitet Medicinska fakulteten, Umea, Sweden
³Department of Community Medicine and Rehabilitation, Physiotherapy, Umea University, Department of Clinical Science, Psychiatry, Umea University, Umea, Sweden
⁴Department of Health Sciences, Luleå University of Technology, Lulea, Sweden

**Correspondence to**
Anna Sandsten;
anna.sandsten@umu.se

## ABSTRACT

**Introduction** People diagnosed with anorexia nervosa are offered or compelled to undergo various treatments and therapies. In Sweden, the lack of highly specialised eating disorder services means that people with anorexia nervosa might be cared for in general psychiatric wards. However, there has been limited research attention paid to how this specific context might benefit or harm persons with anorexia nervosa. There is also limited evidence regarding which form of care is most effective in anorexia nervosa and which type of treatment is optimal in general psychiatric inpatient care. Our intention is to highlight personal and professional perspectives of care for patients with anorexia nervosa in this context, with a focus on person-centred care, gender and power. This protocol describes an integrative literature review aiming to synthesise existing knowledge on general psychiatric inpatient care for persons with anorexia nervosa.

**Methods and analysis** The study will be an integrative literature review following Whittemore and Knafl. We will search the academic databases PubMed, CINAHL and PsycInfo in accordance with a specific search strategy determined in collaboration with a librarian with expertise on systematic reviews. Data will be analysed using a constant comparison method. Persons with their own experience of anorexia nervosa in general psychiatric inpatient care will be involved as coresearchers.

**Ethics and dissemination** As this is a literature review, formal ethical review is not required. The findings will be disseminated through peer-reviewed publication and presented at conferences related to this field.

**Amendments** If we need to amend this protocol, we will give the date of each amendment, describe the change and give the rationale in this section.

**PROSPERO registration number** CRD42023426095.

## STRENGTHS AND LIMITATIONS OF THIS STUDY

⇒ This review will provide knowledge of the content of nursing in general psychiatric inpatient care for persons with anorexia nervosa.

⇒ Persons with their own experiences of anorexia nervosa in general inpatient care will contribute their knowledge and experience as coresearchers and through reading and commenting on the article.

⇒ The publication of this protocol ensures transparency, and evaluation by independent reviewers of all studies during screening and selection reduces the risk of bias in the selection of included studies.

⇒ An integrative literature review combines findings from quantitative and qualitative research which ensures full coverage of research knowledge in the field.

⇒ A limitation is that we exclude studies in languages other than Swedish and English which may lead to potentially relevant studies not being included.

## INTRODUCTION

Anorexia nervosa (AN) is reported worldwide among people of all genders and ages. The incidence among younger persons (aged<15) has increased in recent decades, but the reason for this remains unclear and requires further research.[1] Quality of life for those having AN can be very low, and severe long-term AN significantly increases the risk of premature death due to physical stress or suicide.[2] The psychological foundation of AN can be described by feelings of self-criticism, low self-esteem, isolation, captivity in the disease and hopelessness about the possibility of a good life.[3] On the one hand, AN is a psychiatric diagnosis with a large negative impact on social ability,[4] but on the other hand, it can be experienced as psychologically meaningful. For example, strict rules regarding diet and activity can create structure and predictability in everyday life. Furthermore, AN can function as a strategy to avoid difficult feelings, and can contribute to the experience of control and inner strength.[5 6]

Persons who can no longer manage to cope with their psychiatric illness at home are able to receive psychiatric inpatient care at a hospital.[7] However, the lack of highly specialised eating disorder services means that persons with AN might be cared for in general psychiatric wards, which constitute an environment characterised by rules, locked doors and a lack of choice.[8] This can be stressful, but at the same time being in a

hospital can allow feelings of being safe. General psychiatric inpatient care can be voluntary or coercive, and persons experiencing AN can be offered or compelled to undergo various treatments and therapies.[7]

The organisation of psychiatric services differs worldwide. Inpatient care may be provided in large mental health institutions or in community or general hospital settings. For the purpose of this review, we understand general psychiatric inpatient care to encompass any 24-hour service providing psychiatric care and residence for people with various psychiatric conditions. General psychiatric inpatient care in psychiatry typically means that persons with different diagnoses are cared for in the same place. This leads to a mix of persons with different needs[8]—for example, a person experiencing a manic episode may need reduced stimuli at the same time as a person with depression may need support in finding meaningful activities, and a person with AN may need support in reducing their activity level. Staff describe difficulties in switching between patients' different needs,[9] and wish for strategies and resources for a meaningful care.[10] General psychiatric inpatient care can be understood as a closed system which has specific ideas and expectations of both staff and patients.[11] Staff describe difficulties in maintaining relationships with persons with AN who experience positive aspects of the disease, leading to ambivalent feelings towards care.[12 13] Perceiving a person as 'difficult to treat' can inspire feelings of insecurity in staff. Uncertainty may lead to an increased tendency to rely on guidelines in the care of persons with AN, rather than finding individualised solutions.[14]

According to the National Institute for Health and Care Excellence,[15] treatment for AN should focus on psychoeducation, monitoring of weight, mental and physical health, and risk factors. Care should also be multidisciplinary and involve family members. These recommendations highlight that helping persons with AN to reach a healthy body weight is a key goal, and that weight gain is key in supporting other psychological, physical and quality of life changes that are needed for improvement or recovery. In person-centred care, it is important to empower the patient to have an active role in their own care, to create good relationships and to see the whole person beyond the disease.[16] In recovery-oriented care, the patient owns their narratives in order to create security and enable them to use dialogue to express their expectations and wishes regarding the care and the goals of the care.[11] To enable a turning point towards recovery for persons with AN, it is of great importance for the care to focus on both physical and psychological well-being.[17] Personal recovery from AN is characterised by being seen and understood as an individual, by receiving support in finding meaning in life beyond illness, by feeling hope for recovery and by regaining control and assuming responsibility over life.[18]

Research indicates that care is influenced by societal gender norms, which in turn affect how healthcare professionals interact with patients and interpret patient complaints. Consequently, these gendered expectations may result in patients receiving varying examinations, treatments and medications without sufficient medical justification.[19] This study will summarise the care provided to individuals with AN within the setting of general psychiatric inpatient care. Emphasis will be placed on examining the content of care and the outcomes of care, exploring the discourse surrounding the diagnosis and of the persons having the diagnose. Additionally, the study will shed light on gendered expectations and power imbalances that exist in this particular context.

### Aim
The aim of this study is to synthesise existing knowledge on general psychiatric inpatient care for persons with AN.

### Research questions
The review will seek to answer the following research questions:
- How is the content of care described?
- What is the focus of these descriptions?
- Opportunities/limitations?
- What outcomes are described?
- How are persons experiencing AN described?
- How is AN described?
- Which factors influence care for persons with AN?
- Are gender or power perspectives included in studies?
- How is person-centred care described?

## METHODS AND ANALYSIS
### Study design
This will be an integrative literature review conducted according to the method of Whittemore and Knafl.[20] The purpose of using this method is to cover a large number of studies and to avoid excluding any research on the basis of method.

### Inclusion and exclusion criteria
The studies included in the review will be restricted to those where the setting is general psychiatric inpatient care. Studies describing general psychiatric care for persons of all ages with AN will be included. We will also include studies that describe interventions in general psychiatric inpatient care for persons with AN, and that have a patient, relative and/or staff perspective. Only studies that are peer-reviewed and published in scientific journals will be included. Language of publication will be restricted to English and Swedish, due to the authors' knowledge of languages. Studies with a focus on eating disorders in general will be excluded.

### Information sources
The literature search will start in the spring of 2023, and we will aim for the study to be finished in January of 2024. Based on the specific purpose of the study and its inclusion and exclusion criteria, we will systematically search for knowledge and understanding in studies published in academic databases.[20 21] We will use medical subject headings (MeSH),[22] CINAHL headings and text words related

**Table 1** Draft PubMed search

| Database: PubMed Date: 3 July 2023 | Search terms | Limitations | Results |
|---|---|---|---|
| #1 | anorexia nervosa (MeSH Terms) OR anorexia(Title/Abstract) | | 36 701 |
| #2 | inpatients (MeSH Terms) OR hospitalization (MeSH Terms) OR inpatient care(Title/Abstract) | | 312 995 |
| #3 | #1 AND #2 | Full text, English, Swedish | 991 |

to AN and inpatient care. Databases used in the search will be PubMed, CINAHL and PsycINFO. We will also hand-search the reference lists of research and review articles to make sure that all relevant material has been captured.

## Search strategy

The key concepts in this study are anorexia nervosa and inpatient care, and so the search terms will be 'anorexia nervosa', 'anorexia', 'inpatients', 'inpatient care' and 'hospitalisation'. These search terms are combined using the Boolean operators AND and OR.[22] A block search will be performed with search terms in the form of MeSH-terms,[22] CINAHL headings and free text. The literature search will be limited to peer-reviewed articles published in English. No study design or time limitations will be imposed on the search, and similar searches will be performed in all databases (table 1). Any duplicates will be removed. A Preferred Reporting Items for Systematic reviews and Meta-Analyses flow diagram (figure 1) will be used to organise and summarise the search process.[23] Finally, we will circulate a bibliography of the included articles to the systematic review team. The literature search strategy has been developed in collaboration with a librarian with expertise in systematic reviews at Luleå University of Technology.

## Data management and quality appraisal

The literature search results will be uploaded to Rayyan, an Internet-based software programme that can be used to facilitate collaboration among reviewers during the process of selecting studies.[24] To appraise the methodological quality of studies with different types of methods (ie, qualitative, quantitative and mixed-methods studies), the Mixed Methods Appraisal Tool (MMAT)[25] will be used. The MMAT guides users through four steps: (1) the screening questions are applied to all studies, (2) for each relevant study, the type of design is determined and the corresponding criteria are used to appraise the study's quality, (3) two independent reviewers conduct the appraisal process and (4) an overall quality score is determined for each study.[25] Because of the aim of this review, no studies will be excluded due to quality, instead quality will be visualised and reflected on in the analysis section.

## Selection process

Two authors will independently examine the collected studies using the same questionnaire (table 2). Duplicates

will be removed at this stage. We will obtain full reports for all the titles that appear to meet the inclusion criteria or where there is any uncertainty, and these will be reviewed according to the matrix method.[26] First, we will read through the abstract, and then review the study to determine if it corresponds to the inclusion and exclusion criteria. If so, a copy of the article will be downloaded and sorted in a folder. Studies not meeting the inclusion criteria will be removed, and the reasons for excluding studies will be recorded. We will seek additional information from the study authors when necessary to resolve questions about eligibility. Any disagreement between the authors will be solved by discussion or by a third author. Neither of the review authors will be blind to the journal titles, study authors or institutions.

## Data extraction, analysis and synthesis

Data will be analysed using a constant comparison method as described by Whittemore and Knafl.[20] The analysis is performed in four steps: (1) data reduction, (2) data overview, (3) data comparison and conclusions and (4) verification of data. In the initial step of the analysis, the selected articles are repeatedly read for an overall picture of these content. Based on affiliation regarding focus and content, the articles are then divided into different groups. Answers to the study's research questions are marked with a colour and coded with a number, making it easier to trace the data back to the primary sources. In the second step, the answers are sorted into preliminary subcategories and compiled into a matrix (a data overview) for a better overview of the analysed data. In the third step, new patterns and relationships are identified both within and outside primary data sources, which leads to new subcategories. In the fourth step of the analysis, the data are summarised and verified in accordance with Whittemore and Knafl.[20]

When using this method, the goal of the data analysis is a thorough and impartial interpretation of the primary source that leads to new understanding and broader knowledge in the area studied. We will also aim to investigate whether a gender and power perspective was used in the studies. Discrepancies will be solved through discussion, and we will contact the study authors to resolve any uncertainties. Data extraction will be carried out by the first author and verified by the other authors in order to reduce bias and reduce errors in data extraction. In line with Whittemore and Knafl,[20] we will provide a table summarising all published studies included in the

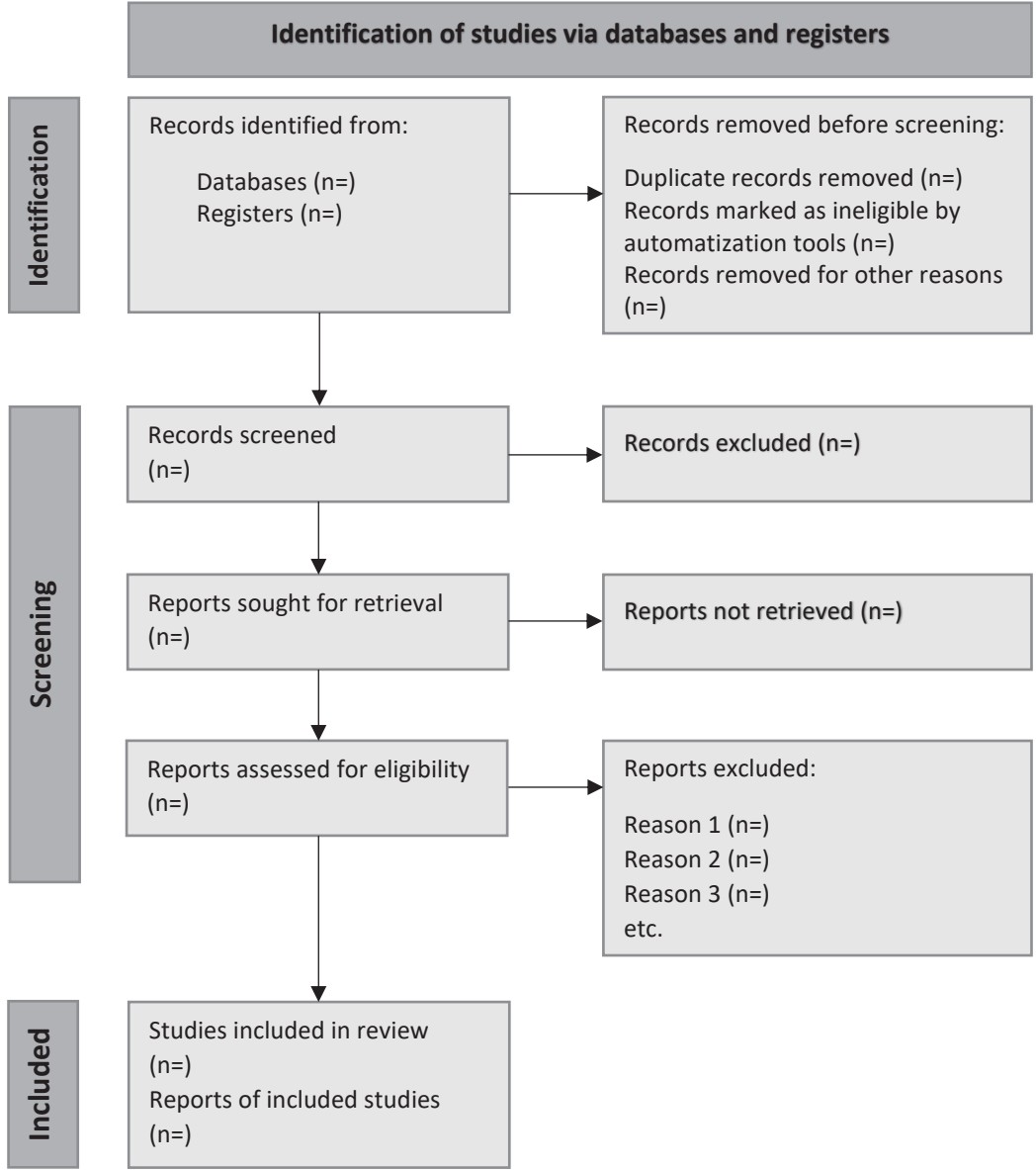

**Identification of studies via databases and registers**

**Identification**

Records identified from:

 Databases (n=)
 Registers (n=)

Records removed before screening:

Duplicate records removed (n=)
Records marked as ineligible by automatization tools (n=)
Records removed for other reasons (n=)

**Screening**

Records screened
(n=)

Records excluded (n=)

Reports sought for retrieval
(n=)

Reports not retrieved (n=)

Reports assessed for eligibility
(n=)

Reports excluded:

Reason 1 (n=)
Reason 2 (n=)
Reason 3 (n=)
etc.

**Included**

Studies included in review
(n=)
Reports of included studies
(n=)

**Figure 1** PRISMA flow diagram for systematic reviews. PRISMA, Preferred Reporting Items for Systematic reviews and Meta-Analyses.

review, comparing their design, population, reference standard, data set size, data-processing methods and diagnostic accuracy. Using this method, we will systematically summarise the narrative and present the synthesis in text and tables to explain the findings of the studies. By doing this, the findings will be explored and summarised both within and between the studies.

### Patient and public involvement

Patients have a right to have an input in research, as this can reduce power imbalances, bring a lived-experience perspective and increase relevance and transparency in research.[27] To broaden the creation of knowledge and to create space for the patient perspective, we will involve a person with own experiences of general psychiatric inpatient care for AN in the research process. The level of involvement will be as coresearcher in study selection, quality appraisal, data extraction, analysis and synthesis.

### DISCUSSION

This review will summarise the content of care for persons with AN in general psychiatric inpatient care. By doing this, we will identify research gaps that can inform further research on how to best create opportunities for personal recovery for these patients.

A strength of this study is that it includes all kinds of care (nursing, psychological, medicine, etc) for persons with AN in this context. This will give an overall picture

**Table 2** Screening and extraction form

| Level 1: Title and abstract review | |
| --- | --- |
| 1. Is this title written in English or Swedish? | Yes/no |
| 2. Does it seem to address AN in general psychiatric inpatient care? | Yes/no |
| Level 2: Full-text review | |
| 1. Is there any reason this article should be excluded? | |
| If yes:<br>What is the reason for excluding? | 1. Not in English/Swedish<br>2. Not about AN in general psychiatric inpatient care<br>3. No full text available despite efforts to retrieve |
| If no:<br>What is the bibliographical information? | 1. Author/s<br>2. Title<br>3. Country<br>4. Publication year |
| What type of publications? | 1. Theoretical (commentary/viewpoint/letter)<br>2. Empirical (qualitative/quantitative/case report) |

AN, anorexia nervosa.

of the content of care and reduce bias in interpretation of different types of care. One limitation is that we might miss articles due to our search strategy—we will only use three data bases, and there might be relevant articles in a data base that we did not search. Moreover, articles might be missed as some key search terms might not be included in their title or abstract. Despite this, this study will highlight what general psychiatric inpatient care has to offer persons with AN today. The results from this review will provide an insight into an under-researched area.

**Contributors** All authors contributed to the conception of the review protocol and the study design of the review. AS and SG developed the search strategy together with a librarian and prepared supplementary materials. AS led the writing of the manuscript, incorporated feedback from SG, G-ME-L, B-ML and MS through multiple drafts. AS and SG prepared correspondence with the publisher. AS, SG, G-ME-L, B-ML and MS contributed to the final manuscript and agreed with its publication.

**Funding** The authors have not declared a specific grant for this research from any funding agency in the public, commercial or not-for-profit sectors.

**Competing interests** None declared.

**Patient and public involvement** Patients and/or the public were not involved in the design, or conduct, or reporting or dissemination plans of this research.

**Patient consent for publication** Not required.

**Provenance and peer review** Not commissioned; externally peer reviewed.

**ORCID iDs**
Anna Sandsten http://orcid.org/0009-0008-6568-098X
Maria Strömbäck http://orcid.org/0000-0003-2597-2466
Sebastian Gabrielsson http://orcid.org/0000-0002-1624-1795

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
