## [Reviewer comments · BMJ Open]

ARTICLE DETAILS

TITLE (PROVISIONAL)	General psychiatric inpatient care for persons with anorexia nervosa - an integrative literature review protocol
AUTHORS	Sandsten, Anna; Lindgren, Britt-Marie; Strömbäck, Maria; Ejneborn-Looi, Git-Marie; Gabriellsson, Sebastian

VERSION 1 – REVIEW

REVIEWER	Hammerle, Florian Universitätsmedizin Mainz, Klinik und Poliklinik für Kinder- und Jugendpsychiatrie und -psychotherapie
REVIEW RETURNED	10-Jul-2023

GENERAL COMMENTS	Thank you very much for the opportunity to review the manuscript “General psychiatric inpatient care for persona with anorexia nervosa - an integrative literature review protocol” describing a study protocol of an integrative literature review study. The topic and the protocol is very interesting – I think this might contribute to synthesize and integrate the developments in the field of inpatient care for patients with AN. The manuscript seems sound and well written. I just have to minor issues: 1. It is not clear if you included a relevant period of time for your literature search (e.g. 2000-2023). Please include a sentence to the used period of time or, respectively include that no time limitation was used in the manuscript (e.g. page 5)2. Another minor concern is the description of the participatory design. I very much appreciate your choice to include patients in the study. Could you provide more information how/at which stages of the study patients are involved? E.g. where patients involved in the study design or are they included in the drafting of the results paper? I look forward to the results. I wish you success.
--

REVIEWER	Barrett, Barbara KCL
REVIEW RETURNED	24-Jul-2023

GENERAL COMMENTS	Thanks for the opportunity to review this protocol. This is a very important area and I think that the integrative review approach is appropriate.
--

	I suggest that you give some greater clarity to the definition of general inpatient psychiatry, as I think that the definition of it is likely to change between countries.
--	---

VERSION 1 – AUTHOR RESPONSE

Reviewer 1:

1. It is not clear if you included a relevant period of time for your literature search (e.g. 2000-2023). Please include a sentence to the used period of time or, respectively include that no time limitation was used in the manuscript (e.g. page 5)

No time limitation is used, this is now clarified, thank you.

2. Another minor concern is the description of the participatory design. I very much appreciate your choice to include patients in the study. Could you provide more information how/at which stages of the study patients are involved? E.g. where patients involved in the study design or are they included in the drafting of the results paper?

We have now added more specific information about this under the heading “patient and public involvement”.

Reviewer 2:

1. I suggest that you give some greater clarity to the definition of general inpatient psychiatry, as I think that the definition of it is likely to change between countries.

Good point, we have now added a more descriptive text on how we understand general inpatient psychiatry in the context of this study.